# Danofloxacin Treatment Alters the Diversity and Resistome Profile of Gut Microbiota in Calves

**DOI:** 10.3390/microorganisms9102023

**Published:** 2021-09-24

**Authors:** Ashenafi Feyisa Beyi, Debora Brito-Goulart, Tyler Hawbecker, Clare Slagel, Brandon Ruddell, Alan Hassall, Renee Dewell, Grant Dewell, Orhan Sahin, Qijing Zhang, Paul J. Plummer

**Affiliations:** 1Department of Veterinary Microbiology and Preventative Medicine, College of Veterinary Medicine, Iowa State University, Ames, IA 50011, USA; afbeyi@iastate.edu (A.F.B.); dgoulart@iastate.edu (D.B.-G.); bruddell@iastate.edu (B.R.); zhang123@iastate.edu (Q.Z.); 2College of Veterinary Medicine, Iowa State University, Ames, IA 50011, USA; tjh3@iastate.edu (T.H.); clareslagel@gmail.com (C.S.); 3Department of Veterinary Diagnostic and Production Animal Medicine, College of Veterinary Medicine, Iowa State University, Ames, IA 50011, USA; ahassall@iastate.edu (A.H.); gdewell@iastate.edu (G.D.); osahin@iastate.edu (O.S.); 4Center for Food Security and Public Health, College of Veterinary Medicine, Iowa State University, Ames, IA 50011, USA; rdewell@iastate.edu; 5National Institute of Antimicrobial Resistance Research and Education, Iowa State University, Ames, IA 50010, USA

**Keywords:** antimicrobial resistance genes (ARGs), bovine calves, bovine respiratory disease, *Campylobacter*, danofloxacin/fluoroquinolones, fecal microbiota, resistome

## Abstract

Fluoroquinolones, such as danofloxacin, are used to control bovine respiratory disease complex in beef cattle; however, little is known about their effects on gut microbiota and resistome. The objectives were to evaluate the effect of subcutaneously administered danofloxacin on gut microbiota and resistome, and the composition of *Campylobacter* in calves. Twenty calves were injected with a single dose of danofloxacin, and ten calves were kept as a control. The effects of danofloxacin on microbiota and the resistome were assessed using 16S rRNA sequencing, quantitative real-time PCR, and metagenomic Hi-C ProxiMeta. Alpha and beta diversities were significantly different (*p* < 0.05) between pre-and post-treatment samples, and the compositions of several bacterial taxa shifted. The patterns of association between the compositions of *Campylobacter* and other genera were affected by danofloxacin. Antimicrobial resistance genes (ARGs) conferring resistance to five antibiotics were identified with their respective reservoirs. Following the treatment, some ARGs (e.g., *ant*9, *tet*40, *tet*W) increased in frequencies and host ranges, suggesting initiation of horizontal gene transfer, and new ARGs (*aac*6, *erm*F, *tet*L, *tet*X) were detected in the post-treatment samples. In conclusion, danofloxacin induced alterations of gut microbiota and selection and enrichment of resistance genes even against antibiotics that are unrelated to danofloxacin.

## 1. Introduction

The rise in the incidence of antimicrobial resistance has become the intractable challenge of public health in the 21st century [1]. Use of antimicrobials in the agricultural sector, especially in the livestock industries, has been identified as a contributor toward the acceleration of the selection and spread of antimicrobial-resistant bacterial strains. This makes livestock farms a potential source of resistant strains and antimicrobial resistance genes (ARGs) [2,3,4]. Gut microbiota is considered an additional “organ” for animals because of its beneficial effects on the physiology, metabolism, immunology, digestion, and nutrient uptake of the host [5,6]. However, it can also harbor ARGs, which may spread to humans through the food chain or environmental contamination [1,4,7,8]. This situation demands a clear understanding of the variables involved in the emergence and maintenance of antimicrobial-resistant bacterial strains and genes in livestock farms and a “One Health” approach to curb their spread to environments and humans.

Fluoroquinolone drugs, such as danofloxacin and enrofloxacin, are frequently used to prevent and control the Bovine Respiratory Disease (BRD) complex in U.S. cattle populations [9,10,11]. BRD is the most significant disease of young cattle in North America. Furthermore, it accounts for 70% to 80% of all feedlot morbidity and 40% to 50% of all mortality in the U.S. feedlots [11]. In 2011, a feedlot survey conducted by the National Animal Health Monitoring System (NAHMS) estimated 16.2% of cattle placed in feedlots showed signs of respiratory disease at some point during their stay on feedlots. Most of these animals (87.5%) were treated with antibiotics, of which 99.0% of the treated calves were injected with antibiotics [12]. Metaphylaxis, mass medication to reduce the incidence of a disease in a group of animals, has also been a well-documented, widely practiced strategy to decrease morbidity and mortality associated with BRD in high-risk cattle [11]. A survey conducted on feedlots by NAHMS in 2011 estimated that 21.3% of all animals entering the feedlot underwent metaphylaxis [12]. BRD is a significant health problem for dairy cattle as well. In 2007, NAHMS surveyed dairies in 17 states and estimated that 12.4% of pre-weaned heifer calves had been diagnosed with respiratory disease, and 5.5% were administered antibiotics [13,14]. BRD incidence has demonstrated seasonal variations and is typically increased in the fall season, which often coincides with the shipping of calves from their farm/ranch origin to the feedlot as well as inclement weather [15]. The Food and Drug Administration (FDA) reports that sales and distribution of fluoroquinolones as well as other medically important drugs have increased in recent years [16]. Fluoroquinolones, used for the control of BRD and individual animal treatment, accounted for 1.8% of the total regimens of antimicrobial drugs used in 22 U.S. beef feedyards in the years 2016 and 2017, which was moderately high compared to some other antimicrobials used for the same purposes (ranging from 0.11% (sulfonamide) to 9.9% (macrolide)) [17].

Ciprofloxacin, a fluoroquinolone antibiotic with the same principal agent as that of animal formulations, is commonly used to treat various human infections, including gastroenteritis caused by *Campylobacter* infection [18,19]. The increased incidence of ciprofloxacin resistance among foodborne pathogens has become a serious threat to public health, which has led to the first removal of fluoroquinolones from use on poultry farms in the U.S. in 2005 [20]. Despite the withdrawal, the trend of fluoroquinolone resistance in *Campylobacter* species has continued to increase in the last two decades (https://www.cdc.gov/DrugResistance/Biggest-Threats.html#cam, accessed on 14 August 2020). It has been suggested that the rise in ciprofloxacin resistance in *Campylobacter* in humans may, at least in part, be associated with use of danofloxacin and enrofloxacin in other livestock [21,22]. Beyond inducing antibiotic resistance in specific bacterial pathogens, these antibiotics exert selection pressure on gut microbiota that may lead to its alteration and enhance the spread of ARGs within bacterial communities [23,24,25].

Perturbation of gut microbiota following exposure to antibiotics has been a well-recorded phenomenon in monogastric animals, including humans [26,27,28,29,30,31]. However, only limited information is available in cattle, and the available data demonstrate inconsistent results and conclusions among studies. In one study, intramuscularly administered penicillin modified the structure of fecal microbiota in calves [32]. Conversely, another group of researchers reported only minor alterations in community structure and the absence of changes in fecal microbial diversity in steers that received single and multiple doses of enrofloxacin subcutaneously [33]. In agreement with the 2018 study, comparisons of the fecal microbiota of calves with and without a history of antimicrobial exposure on beef farms were not different [34]. However, *E. coli,* used as an indicator organism for the impacts of antibiotics on gut bacteria, showed transient resistance (i.e., a temporary increase in minimum inhibitory concentration) [33], and increased resistance to multiple unrelated antimicrobial agents [32]. In general, the inconsistency among the studies could partly be attributed to differences in their study designs and methods as well as the inherent complexity within the gut microbiota [35].

The objective of this study was to assess the impacts of danofloxacin administered subcutaneously to bovine calves on the fecal microbiota, *Campylobacter*, and resistome (i.e., all detectable ARGs within a given sample). We hypothesized that infusion of danofloxacin and its metabolite, ciprofloxacin, from plasma into the intestine following a single-dose subcutaneous injection of danofloxacin will exert selective pressure on gut microbiota and alter its diversity and the resistome profile. In this study, we sequenced the 16S rRNA gene to compare the compositions and structures of fecal microbiota before and after the treatment and to assess changes in the compositions of *Campylobacter*. Additionally, we employed a novel proximity ligation-guided metagenomics approach (ProxiMeta) that has the advantage of linking the ARGs with their microbial reservoirs over other metagenomic methods [36], which enabled us to track changes in antibiotic-resistant determinants as driven by danofloxacin administration. Quantitative real-time PCR (qPCR) was employed to assess the impacts of danofloxacin on the dynamics of selected ARGs.

## 2. Results

### 2.1. Results of 16S rRNA Gene Analysis

The total number of reads in our samples was 10,999,180, with an average of 52,377 reads per sample (SD 12,721, range 5179–124,738, and median 51,698 reads per sample). A total of 6628 bacterial Operational Taxonomic Units (OTUs) were identified in 210 samples, which were classified into 298 genera (mean = 91 genera, SD = 7, and range = 68–116 per sample), 35 classes (mean = 23 classes, SD = 2, and range = 16–28 per sample), and 19 phyla (mean = 13 phyla, SD = 1, and range = 9–17). A random subsample of 11,500 reads per sample was used to normalize sequence numbers for the computation of alpha and beta diversity metrics; one sample with 5179 reads was excluded from the analyses due to insufficient depth.

#### 2.1.1. Bacterial Phyla by Sampling Days

The relative abundances of phyla detected in this study were compared by the sampling days (Appendix A). Phyla with the highest relative abundance in the first sampling day, two days after the calves arrived at the animal study facility, included *Firmicutes* (relative abundance = 51.93%), *Bacteroidetes* (36.13%), *Proteobacteria* (3.00%), *Tenericutes* (2.18%), *Spirochaetes* (1.94%), *Actinobacteria* (1.33%), and *Verrucomicrobia* (1.03%). The relative abundances of the phyla varied over the subsequent sampling days during the four weeks of the study period; however, *Firmicutes* and *Bacteroidetes* remained dominant phyla throughout that time.

#### 2.1.2. Alpha Diversities

The sequencing depth was evaluated using alpha rarefaction curves (figures not shown); for all considered alpha diversity metrics, the sequencing depth was sufficient to reveal rare microbes in the samples. Calves in group C that were challenged with *Mannheimia haemolytica* exhibited mild to moderate signs of BRD. However, microbial compositions and diversities were not significantly different from group B, the group that was not challenged with *M. haemolytica* but injected with danofloxacin subcutaneously. Thus, these two groups that received danofloxacin were collapsed together into one group, named the treatment group, for the comparison of microbial evenness and richness before and after the treatment. Based on the analyses, observed OTUs measuring the microbiota richness showed a significant change between pre- and post-treatment samples (*p* < 0.05). Similarly, microbiota evenness measured by the Shannon index was different between pre- and post-treatment samples (*p* < 0.05, Figure 1). Strikingly, both microbial richness and evenness significantly increased following danofloxacin administration, as shown in Figure 1.

Since the fecal samples were collected from the study calves over 28 days, there was an anticipation that the fecal microbiota would change over time because the calves were around two months old at the beginning of the study, the age at which microbial diversity and structure alter fast. Thus, linear mixed-effects models were run using QIIME2 to discriminate between microbial changes attributed to age from those due to danofloxacin exposure. Accordingly, the effect of time was not significant for either measure in the control group. In contrast, the time had a marginally significant influence (observed OTUs, *p* = 0.064 and Shannon, *p* = 0.091) in the treatment group, indicating the influence of danofloxacin on the microbial compositions and diversities was more prominent than the impact of the time in this study.

#### 2.1.3. Beta Diversities

Beta-diversity was measured by the Bray–Curtis dissimilarity index, which can be visualized in the NMDS plot displayed in Figure 2. Gut microbiota shifted after the treatment, as shown in the figure by a clustering difference between pre- and post-treatment microbial communities, and the community structures were significantly different (*p* < 0.05).

#### 2.1.4. Comparisons of Compositions of Bacterial Classes among Study Calf Groups

The relative abundances of bacterial taxa were compared among the calf groups using the Kruskal–Wallis and pairwise Dunn tests (Appendix A). For many of the bacterial classes, a significant difference (*p* < 0.05) was observed between the control group (Group A) and the treatment groups (Group B or C, or both). For instance, *Methanobacteria, Thermoplasmata, Betaproteobacteria, Epsilonproteobacteria,* and *Mollicutes* significantly increased in the treatment groups, whereas *Bacilli, Planctomycetia,* and *Synergistia* decreased. Some bacterial classes, such as *Bacteroidia, Cyanobacteria-4C0d-2,* and *Clostridia* did not show significant changes. Comparisons between the treatment groups, which were BRD-induced Group C against non-BRD-induced Group B, had shown significant differences for certain bacterial classes such as *Actinobacteria, Planctomycetia, Betaproteobacteria,* and *Tenericutes- RF3*.

#### 2.1.5. Comparisons of Relative Abundance of Bacterial Genera between Pre- and Post-Treatment Communities

After the classification of the reads at a minimum of 97% similarity into OTUs using QIIME 1.9, a total of 4770 OTUs were obtained. Two hundred and sixty-five of the OTUs in the treatment group increased, and 211 of them decreased after danofloxacin injection at FDR *p* ≤ 0.01 (379 and 319 at FDR *p* ≤ 0.05, respectively). Furthermore, the relative abundance of genera in pre- and post-treatment samples were compared using the Wilcoxon signed-rank test and Boruta algorithm/random forest models. Out of 228 genera with relative abundance ≥ 0.01%, 66 of them had significantly shifted after the treatment as indicated by the Wilcoxon signed-rank test (*p* < 0.05, Table 1) and 57 genera after the *p*-value was adjusted by FDR (*p* < 0.05). These genera fell into 12 phyla, with 31 of them in *Firmicutes*, 13 in *Proteobacteria*, 8 in *Bacteroidetes*, 3 in *Actinobacteria*, 3 in *Spirochaetes*, 2 in *Elusimicrobia*, and the rest in the other six phyla.

For the classification of the microbial communities into pre- and post-treatment, out of 298 genera, 44 of them were confirmed as important attributes by the Boruta algorithm, meaning the treatment affected the proportions of these genera significantly. The feeding of these important genera to the random forest model resulted in an out of bag error of 2.06%, accuracy for the training sub-data of 100% (95% CI: 96.27–100 %), and an accuracy rate for the test sub-data of 100% (95% CI: 91.59–100 %). These 44 genera, classifying the microbial communities into pre- and post-treatment communities, were ranked by the random forest model based on their importance, as depicted in Figure 3. Four *Proteobacteria* genera (*Succinivibrio*, RF32_g_unclassified, *Gallibacterium*, *Enterobacteriaceae*_g_unclassified) and *Bacteriaodetes*_g_5.7N15 were the most important. Interestingly, 37 genera were identified as having a significant association with or predictors of pre/post-treatment status by both the Wilcoxon signed-rank test and the Boruta/random forest method.

Furthermore, the number of genera per sample 24 h after the treatment was significantly lower (*p* < 0.05) in the treatment group than in the control group (Figure 4). For groups B and C, the counts were significantly reduced at 24 h after the treatment compared with one day earlier sampling or four or seven days after the treatment (i.e., sampling days 19 vs. 18, 19 vs. 22, 19 vs. 28, *p* < 0.05; day 18 was the treatment day). In contrast, none of them were significant in the control group.

### 2.2. The Compositions of Campylobacter

#### 2.2.1. Comparison by Sampling Days

The relative abundances of *Campylobacter* in samples collected on Day 2 through Day 28 were 0.03%, 0.03%, 0.35%, 0.72%, 0.04%, 0.31%, and 0.76% in the pooled data of the combined treatment groups (groups C and B), respectively. Its composition on Day 19 (24 h after danofloxacin administration) was significantly lower than the composition on Day 18, as well as than those on the last two sampling days. Figure 5 depicts the proportions of *Campylobacter* in the samples collected at seven sampling times compared by the Kruskal–Wallis non-parametric and its post hoc tests. *Campylobacter* level dropped sharply and significantly 24 h after the treatment; however, it returned to the pre-treatment baseline after three days (i.e., no significant difference between the proportions on Day 18 and Day 22, *p* > 0.05). In contrast, the proportions of *Campylobacter* did not show significant variations among sampling days except for a slight rise in the last sampling day in the control group (results not shown).

#### 2.2.2. Correlation with Other Genera

The relative abundance of *Campylobacter* was compared with other genera using the Spearman rank correlation test (Appendix A). Twenty-one genera had a significant correlation with *Campylobacter* in the pooled data (all groups combined), of which ten of them had a negative correlation. Seven of the eight *Bacteroidetes* genera that significantly correlated with *Campylobacter* had negative correlation coefficients. In contrast, most of the genera in *Firmicutes* and *Proteobacteria* had a positive association with the relative abundance of *Campylobacter*.

On the other hand, pre- and post-treatment data were separately analyzed to assess if the treatment had an effect on the pattern of correlation between *Campylobacter* and other genera. Seven and eleven genera in pre- and post-treatment samples had a significant correlation with *Campylobacter*, respectively. Interestingly, pre- and post-treatment samples shared only one genus in common that had a significant association with *Campylobacter*, which was an unclassified genus in the family of *Desulfovibrionaceae*. This result suggests that danofloxacin exposure altered the correlation pattern between bacterial taxa.

#### 2.2.3. Prediction of Important Genera by Random Forests

Using the Boruta algorithm, out of 298 genera, nine of them were confirmed as important attributes in the combined data of all groups. Based on the random forest model, these nine genera explained 46.01% of the variability in the relative abundance of *Campylobacter* with a very low mean squared error (i.e., 0.059), indicating the best model fitness. Except for two, these genera had significant Spearman rank correlation coefficients as well. Furthermore, genera ranked as very important by the random forest model had the highest Spearman correlation coefficients. Additional analyses were performed by dividing data as pre-treatment, post-treatment, and control to check if the treatment would have changed the association patterns between genera. In agreement with the Spearman rank correlation test, there was a difference between pre- and post-treatment samples in both the types and numbers of genera correlated with the compositions of *Campylobacter* in the fecal samples. Figure 6 presents genera that predict the relative abundance of *Campylobacter* in the total, the pre- and post-treatment, and the control samples separately.

### 2.3. Results of Metagenomic Hi-C (ProxiMeta)

To assess the influence of danofloxacin injection on fecal resistome profiles, pooled samples from pre- and post-treatment samples were sequenced using metagenomic Hi-C. Accordingly, genes conferring resistance to tetracycline, aminoglycosides, β-lactams, macrolides, and phenicol were detected in both pre- and post-treatment samples (Table 2). The abundance of tetracycline resistance determinants was the highest with eight and nine different ARG types in pre-and post-treatment pooled samples, respectively. *tet*Q, *tet*W, and *tet*40 showed a dramatic increase both in number of hits and host ranges following the treatment; *tet*B was detected only in the pre-treatment sample, while *tet*L and *tet*X were identified after the treatment samples only. *tet*32 and *tet*O were carried on the same contig. Six unique aminoglycoside ARG types were detected in the samples with *aac*6 only observed in the post-treatment sample. The number of *ant*9 copies increased dramatically following the treatment from 39 to 189 with host range increase from 12 to 23. The rest of the aminoglycoside ARGs showed minor variations between pre-and post-treatment pooled samples. Two or more ARG types were carried on the same contig. Four unique macrolide ARGs were identified, *erm*B being undetected in the post-treatment sample, while the rest of them did not fluctuate both in copy numbers and host ranges between the two sampling time points. *aci* and *cf*X β-lactam ARGs were reported in this study with the latter ARG emerged after the treatment. Only one phenicol resistance gene (*cfr*) was observed, which significantly increased both in the number of hits and reservoirs following the treatment.

Using proximity ligation, the linkage of ARGs with their respective bacterial hosts was also established in the present study. Phyla associated with the current ARGs are presented in Table 3. *Firmicutes* was the leading phylum associated with these ARGs, whereas *Bacteroidetes* appeared to acquire the ARGs following the treatment. At lower bacterial classification level, most of these resistance genes were hosted by class *Clostridium* species, *Eubacterium* species, and *Sharpea azabuensis* DSM_18934 in the pre-treatment samples, and *Clostridium, Eubacterium*, *Prevotella,* and *Ruminococcus* species in the post-treatment samples (Appendix A). Most of the identified ARGs were carried by more than one clusters of bacterial taxa having ≥ 80 % complete genomes; for instance, *tet*W was detected in 12 and 49 clusters in pre-and post-treatment samples, indicating a wide range of distribution. Appendix A presents copy numbers and bacterial taxa (at a lower classification level) associated with each ARG.

### 2.4. Results of qPCR

To validate the Hi-C results, a subset of genes was selected for qPCR testing. Primers targeting *tet*W, *tet*O, *tet*X, *erm*B, and *erm*F were used to compare the changes in their quantity following danofloxacin administration. Comparison of the treatment groups (B and C) with the control group using the non-parametric pairwise Dunn test has revealed that *tet*W increased significantly (adj. *p* < 0.05) in the post-treatment pooled samples from group B compared the control. In contrast, *tet*O was significantly reduced (adj. *p* < 0.05) in group C compared to the control while *erm*F increased significantly in the same group. The change in the quantity of *erm*B was significantly lower in group B than the control group (Table 4).

## 3. Discussion

Danofloxacin is frequently administered to beef calves as prophylaxis and metaphylaxis to prevent and control BRD in beef cattle in the U.S. Having a short meat withdrawal time makes it one of the preferred antibiotics used by producers [37]. However, little is known about its unintended side effects on gut microbial compositions, diversities, and resistome. In this study, we investigated the impacts of a single dose of subcutaneously administered danofloxacin on the fecal microbiota, the compositions of *Campylobacter*, and the resistome in calves. Compared to the control group, fecal microbiota was shifted in the treatment groups following danofloxacin injection, as indicated by significant changes in alpha and beta diversities as well as the shift in the relative abundance of bacterial taxa. Additionally, the abundance of *Campylobacter* has been altered in the treatment groups, likely due to the emergence of antibiotic resistance in this genus (data being prepared under a separate manuscript). Furthermore, our study showed enrichment of antibiotic determinants, which suggests that danofloxacin created an environment that might have favored the dominance of resistant strains and/or transfer of ARGs among bacteria. Given that antibiotic use exerts selective pressure on bacteria for the emergence of resistance, these findings are consistent with evidence by many other studies. The emergence of resistant strains and genes in livestock is worrisome since they shed them with feces and contaminate the farm environments, which may eventually reach the human food chain or drinking water. This necessitates capturing the “One health” paradigm to study AMR and minimizing its economic and public health impacts [7,8].

The effect of danofloxacin on bovine gut microbiota has not been previously studied despite being one of the commonly used fluoroquinolone drugs on beef farms. Fluoroquinolones have a bactericidal effect on bacteria by interfering with the DNA coiling during replication [37]. Their mechanisms involve inhibition of bacterial enzyme topoisomerases, such as topoisomerase II (DNA gyrase) in Gram-negative bacteria and topoisomerase IV in Gram-positive bacteria. Danofloxacin is a third-generation fluoroquinolone drug used for the treatment of respiratory disease in pigs and cattle [38]. Parenteral administration of this drug results in a high concentration in the digestive tract, which increases antibiotic exposures of gut microbiota. According to a study performed on dairy cows, the concentration of danofloxacin in tissue samples from the digestive tract of cattle was high. It was more than five times and 19 times the plasma concentration in the colon and mesenteric lymph nodes at 8 h and 12 h post-administration, respectively [38]. The same study indicated a high volume of distribution and prolonged elimination half-life of danofloxacin. Other researchers also demonstrated that marbofloxacin (another fluoroquinolone) doses used to target pathogens in the lungs had a similar effect on the bacterial population in the small intestine [39]. These authors reported 2- to 8-fold higher levels in total marbofloxacin concentrations in the digestive tract than in plasma for over 24 h following intramuscular injection of this drug to young pigs [39]. However, the bactericidal effect of fluoroquinolones in the intestine is diminished because they are highly bound in the intestines/feces, as demonstrated in pigs [39], rats [40], and humans [41]. Despite the reduced bactericidal effect, available evidence shows that fluoroquinolones create selective pressure in the intestine [18]. One such example is the emergence of fluoroquinolone resistance in *Enterobacteriaceae* strains following intramuscular injection of enrofloxacin in pigs [42]. The same study revealed that both orally and parenterally administered enrofloxacin induced development of AMR in selected bacterial species with different levels in pigs [42]; however, the difference in the impacts of fluoroquinolone drugs administered via oral and parenteral routes on gut microbiota needs to be investigated in calves. Nevertheless, there is enough evidence from the literature that complements our findings and shows parenterally administered danofloxacin is capable of enhancing selective pressure on gut microbiota and inducing its perturbation.

The major bacterial phyla identified from the fecal samples collected after two days of the beginning of the study (i.e., baseline data) were *Firmicutes* (51.93%), *Bacteroidetes* (36.13%), *Proteobacteria* (3.00%), *Tenericutes* (2.18%), *Spirochaetes* (1.94%), *Actinobacteria* (1.33%), and *Verrucomicrobia* (1.03%). These findings agree with previous studies in beef cattle [43,44,45,46]. A study conducted in Canada, however, indicated a variation between beef farms with regard to bacterial dominance; some of the farms were characterized by the dominance of *Actinobacteria* and *Proteobacteria,* while the other farms by the dominance of *Firmicutes* [34]. Several factors, such as diets, antibiotic exposure, genetics, management, and environment, contribute to the variation in microbial compositions and diversities between and within farms [28,30,47,48]. In the current study, the calves were originated from the same farm, fed the same diets, and managed the same way under similar environment; and none of them had a history of antibiotic exposure prior the study. Thus, microbial variations attributed to these factors were minimal.

In the present study, danofloxacin increased microbial richness (median OTUs rose from 293 to 313) and diversity (median Shannon index rose from 6.98 to 7.08). These findings contrast with our initial hypothesis that the higher concentration of danofloxacin diffusing into the intestine reduces microbial richness and diversity. Consistent with our original hypothesis, a two-week tetracycline administration reduced gut microbial diversity, changed the abundance of some bacterial taxa, and enhanced the abundance of ARGs, plasmids, and integrons in mouse guts [24]. In agreement with our current finding, other authors reported that diversity of microbiota increased in the piglets that received a single injectable dose of tulathromycin compared to the control, which they considered as “more chaotic” [30]. On the other hand, the numbers of genera in the treatment groups dramatically and significantly reduced 24 h after danofloxacin injection compared to the control group in this study. However, these numbers returned to the pre-treatment level three days later, which may indicate a transient effect of danofloxacin on microbial richness. Furthermore, a significant clustering difference between pre- and post-treatment microbial communities depicted in the NMDS plot (Figure 2) demonstrated the impact of danofloxacin on the structures of microbial communities. In contrast to the present study, Ferguson and colleagues [33] reported only minor changes in community structure and the absence of changes in microbial diversity following enrofloxacin administration in steers. The discrepancy between these two studies could be partly attributed to the difference in the study designs, where we used ten calves per group under a controlled study setup, and they employed six animals per group. However, it is worthwhile to investigate if the pharmacokinetic differences that exist between danofloxacin and enrofloxacin might have contributed to the variations between these two studies [49].

In the present study, we assessed the effect of danofloxacin on the composition of *Campylobacter* and its association with the other genera. We were particularly interested in this genus because of the public health significance that some species belonging to *Campylobacter* have and the rising of antibiotic resistance among them [50]. Twenty-six of the thirty study calves were colonized by *Campylobacter* species that have public health importance (i.e., *C. jejuni*) when they were first enrolled in this study, as indicated by bacterial culture (data not shown). Following the inoculation of the laboratory strains, the proportion of *Campylobacter* (i.e., based on the 16S study) continued to rise until the day danofloxacin was administered to the calves in the treatment groups. The increase in the abundance was not quick as there was no significant difference between the sampling time points 1 and 2 (Day 2 and Day 6, the laboratory strains were administered on Day 4 after the start of the study), and the absence of a difference between the two sampling points could be due to the delay in *Campylobacter* colonization arising from fighting for niches with other microbes [51]. After danofloxacin injection to the treatment calves, the relative abundance of *Campylobacter* sharply reduced in the samples collected after 24 h; however, it rebounded to the pre-treatment baseline after three days (Figure 5), suggesting that resistant strains emerged as a result of the antibiotic exposure. In agreement with this finding, previous studies conducted by our research team reported the emergence of fluoroquinolone resistance in *Campylobacter* strains in chicken intestine as early as 24 h after the initiation of oral enrofloxacin treatment. The proportion of this genus remained unaltered in the control group except for a slight rise in the last sampling day, which strongly suggests that the changes that observed in the treatment groups were driven by danofloxacin administration.

Furthermore, in the current study, we identified 21 genera that had a significant correlation (*p* < 0.05) with *Campylobacter*. Most of them were confirmed to have a correlation with *Campylobacter* by the Boruta important variable selection algorithm and the random forest models, which are robust methods to identify correlation between variables [52,53,54]. Consistent with previous studies, most of the genera in *Bacteroidetes* phylum had a negative correlation and genera in *Firmicutes* and *Proteobacteria* phyla had a positive correlation with *Campylobacter* in the present study [55,56,57]. However, there are some variations with those studies; for instance, we found a negative correlation between *Campylobacter* and *Prevotella* in contrast to the report by Dicksved and colleagues [57]. Understanding the association between gut bacterial taxa has public health implications because gut microbial compositions affect colonization resistance to enteric bacterial infections, including *Campylobacter* [50,56]. For instance, 12 bacterial taxa were shown to have a significant impact on the prevalence and enumeration of *E. coli* O157:H7 in cattle feces [58]. We also found that the genera that had a correlation with *Campylobacter* were different before and after danofloxacin injection, except for one genus (Appendix A). The change in the patterns of association between gut microbiota may have to do with the way different bacterial taxa respond to antibiotics. As indicated in Table 2 and Table 3, some of the bacterial taxa significantly increased or decreased in their compositions following danofloxacin injection, while the other remained unaltered.

The mechanisms by which gut microbiota affect the compositions of *Campylobacter* include direct inhibition, altering the intraluminal milieu, and production of metabolites [50,57,59]. Exploring the relationship between *Campylobacter* and gut microbial compositions can give an insight into how to mitigate the spreading and transmission of this foodborne pathogen to humans. Deliberate alteration of gut microbial composition could help to control the amount of *Campylobacter* shed in cattle feces [58]. Our findings call for further investigation of the identified bacterial taxa with a negative correlation with *Campylobacter,* particularly those genera with high correlation coefficients or high importance in the random forest plots, which could be used as probiotics to curtail the fecal shedding in cattle or other food animals. Several bacteria, mainly *Lactobacillus* species, have been shown to reduce colonization and shedding of *C. jejuni* in poultry [60,61,62]. Brooks and colleagues observed that antibiotic-driven depletion of gut microbiota in a mouse model increased *C. jejuni* colonization, invasion, and severity of gastroenteritis, which may require administration of probiotics following *C. jejuni* infection to ameliorate inflammation and autoimmune disease [63].

In the current study, the metagenomic Hi-C ProxiMeta predicted the presence of ARGs that confer resistance to tetracycline, aminoglycoside, β-lactams, and erythromycin in the fecal samples. The presence of ARGs of these antibiotics in the samples is not a surprise because they are among the most frequently used antibiotics on beef farms for the control and prevention of diseases and as feed additives [17]. According to the FDA report, these antibiotics accounted for 67% (tetracyclines), 12% (β-lactam penicillin), 8% (macrolides including erythromycin), and 5% (aminoglycoside) of the total medically important antibiotics approved for use in food-producing animals and actively marketed in 2019 [64]. Furthermore, based on a recently conducted study on 22 U.S. beef farms, macrolide and tetracycline in-feed use accounted for 59.7% and 19.4% of the total regimens used in 2016 and 2017, respectively [17]. The in-feed ingredient use of these antibiotics in the feedlot production system contributes toward the emergence and spread of resistance; even, in those farms that stopped using these drugs a long time ago, detection of their resistance genes is a common phenomenon [65]. In agreement with our findings, Zaheer and colleagues [43] documented a high prevalence of tetracycline resistance (82%), followed by macrolide (14%) and aminoglycoside resistance (2.2%) in fecal samples from beef cattle.

Fluoroquinolone (in this case danofloxacin) resistance is primarily conferred through a single mutation in *gyr*A gene that is involved in DNA supercoiling [66]. The method we employed in the current study does not detect a single nucleotide change, despite the rebound of *Campylobacter* four days post-treatment, suggesting the emergence of resistant strains, which also complemented by previous trials in our laboratory and in the literature [18,25]. Genes that have been shown to be associated with fluoroquinolone resistance in other bacteria (e.g., *qnr*A, *qnr*B, *qnr*S, etc.) were not detected in the current study. In a previous study that assessed the effect of danofloxacin treatment (IM, for three days) in pigs on the *Campylobacter coli* resistance profile, the MIC of the isolates increased after the treatment and attributed to the mutation of the *gyr*A gene [18].

In this study, antibiotic resistance determinants have increased both in number and types after danofloxacin administration. The detection of new types of ARGs (i.e., *erm*F, *sat*, *tet*L, and *tet*X; Table 2) that were not detected in the pre-treatment samples might trigger a question of why resistance genes unrelated to danofloxacin had emerged after the treatment. We propose two hypotheses supported by literature. (1) An enrichment of bacterial strains resistant to these antibiotics: danofloxacin might have wiped out or weakened susceptible bacterial strains and selected for resistance strains to predominate others. The identified resistance genes might have been present before the treatment at a level below the detection limit; however, the treatment might have enriched the bacterial strains hosting these ARGs to the level where our method was capable of detecting them. Parallel to our observation, the sulfonamide resistance gene (*sulf*2) was enriched following the administration of unassociated antibiotics such as macrolides and tetracycline in cattle [67]. (2) These ARGs might be available at an undetected level before the treatment; however, danofloxacin administration might have favored horizontal gene transfer among genetically related and, probably, unrelated bacteria. Horizontal gene transfer is a common way of acquiring ARGs by bacteria in complex environments, such as in the gut [27,68,69]. Co-transfer of *erm*B and *tet*M in *Streptococcus pyogenes* was reported, which suggested their linkage in individual genetic elements [70]. Colocalization of many ARGs on microbial genomes was found in swine feces and manure; they confer multi-drug resistance to bacteria within a community [71]. In the current study, the exposure to danofloxacin might have enhanced both the dominance of resistant bacteria as well as the transfer of the resistance genes among the microbes in the bacterial communities horizontally [72]. In one study, it was shown that danofloxacin treatment decreased the susceptibility of *C. jejuni* to macrolide (tylosin) [18]. Meanwhile, in the mouse intestinal microbiota, tetracycline treatment enhanced the abundance of antibiotic ARGs (from 307.3 to 1492.7 ppm), plasmids (from 425.4 to 3235.1 ppm), and integrons (from 0.8 to 179.6 ppm) [24].

The ProxiMeta Hi-C metagenomic deconvolution also enabled us to predict bacterial taxa that hosted the ARGs. Previously, using the metagenome-assembled genomes reconstructed by this method, Stalder and colleagues detected an association between ARGs, mobile genetic elements, and host genomes from wastewater samples [36]. In our study, the order *Clostridiales* in class *Clostridia* hosted the largest number of ARGs, where *ant*6, *ant*9, *aph*2, *sat*, *tet*O, *tet*W, *tet*32, *tet*W40, and *tet*44 were hosted by this bacterium. *Sharpea azabuensis* DSM_18934 (in the phylum *Firmicutes*, class *Erysipelotrichia*, and family *Coprobacillaceae*) identified in the pre-treatment samples was found to host *ant*6, *ant*9, *cfr*, and *tet*W. The genus *Sharpea* was detected in 78 of 80 pre-treatment and 59 of 60 post-treatment samples using the 16S rRNA analysis, but it was not identified by the Hi-C metagenomics in the post-treatment samples. *Sharpea azabuensis* is a ruminal bacterium that produces *trans*-11 intermediates from linoleic and linolenic acid [73]. Information about the antimicrobial resistance status or acting as a reservoir of resistant genes is not available for this species. In general, the Hi-C findings are consistent with a longitudinal study conducted in China, where they quantified resistance genes in manure from pigs using real-time PCR (i.e., qPCR) and 16S gene sequencing [68]. The authors reported a strong positive correlation between the dynamics of tetracycline resistance genes (i.e., *tet*M, *tet*O, *tet*Q, and *tet*W) and the relative abundances of certain OTUs of gut associated *Clostridiales*. Furthermore, higher levels of *tet*W and *tet*40 were detected in *Bacteroidetes* following danofloxacin treatment, suggesting the occurrence of ARG transfer horizontally between these *Clostridiales* in this phylum and other bacteria. Previous studies show that bacterial species in this family acquire resistance genes from other bacteria through transposons and plasmids [74,75]. It appears that danofloxacin administration favors expansion of the ARG host ranges. In this study, 11 and 6 ARG types increased and decreased in copy numbers following the treatment, while 12 and 7 of them had their host range increased and decreased, respectively.

Selected ARGs identified by the metagenomic Hi-C method were further quantified using qPCR in the current study. The results of the latter method were consistent with the Hi-C results in that the quantity of *tet*W and *erm*B were altered significantly, while that of *tet*O, *tet*X, and *erm*F were not affected in the post-treatment fecal samples in group B. However, the dramatic increase in *tet*W predicted by the Hi-C method was not measured by qPCR, which might be attributed to one or both of the following reasons: the fecal samples used in the case of Hi-C ProxiMeta was a pooled sample from two calves on the danofloxacin injection day (i.e., Day 18) for pre-treatment sample and four days later (i.e., Day 22) for the post-treatment sample. In the case of qPCR testing, however, DNA extracts from all calves (ten) in group B were pooled together with a different post-treatment day. The pre-treatment sampling day was the same (i.e., Day 18) but the post-treatment sampling day was different (i.e., Day 28). Nevertheless, the prediction of the trends of ARG changes between the well-established gene quantification method (qPCR) and the novel metagenomic Hi-C method is similar, suggesting the robustness of this novel method in predicting the dynamics of antibiotic determinants in complex microbial communities, such as gut or fecal microbiota.

## 4. Materials and Methods

### 4.1. Study Design and Animals

For this study, 30 dairy calves (26 male; 4 female) with predominantly Holstein genetics and no prior history of antibiotic exposure were acquired from a dairy farm located in the state of Iowa. Calves were approximately 8 weeks old at the time of procurement and weighed between 54 kg to 93 kg. Following purchase, they were group-housed in three rooms (ten calves in each room) at the Livestock Infectious Disease Isolation Facility (LIDIF) at Iowa State University (Biosafety level 2) for 28 days. The rooms were maintained at 20–21 °C (68–70 °F), and each of them had an independent airflow system. Upon arrival at the facility, calves were visually examined by veterinarians for signs of disease such as lameness, nasal discharge, dyspnea, obtundation, ophthalmic abnormalities, bloat, or diarrhea. Only calves that appeared healthy were enrolled in the study. Following visual examination, calves were weighed and then randomly assigned to one of three study groups after blocking by their weight. Study groups included: Group A—control, oral inoculation with *C. jejuni*; Group B—oral inoculation with *C. jejuni* and subcutaneous administration of a single dose of danofloxacin; and Group C—oral inoculation with *C. jejuni*, trans-tracheal inoculation with *Mannheimia haemolytica*, and subcutaneous administration of single dose of danofloxacin. Throughout the study course, calves were fed mixed grass hay and a pre-mixed calf starter (Heartland Co-op, Des Moines, IA, USA) and provided with *ad libitum* water. None of these calves showed serious health problems that required additional antibiotic administration during the study period. The procedures in this study were undertaken following prior approved Institutional Animal Care and Use Committee requirements for Iowa State University.

Following a four-day acclimatization period, all calves were orally inoculated with *C. jejuni* (a cocktail of laboratory strains obtained from Texas, Missouri, and Colorado). The inoculum was prepared by combining fresh colonies of *C. jejuni* with Mueller–Hinton (MH) broth. For each calf, 60 mL of *C. jejuni* suspension (~10 × 10^9^ CFU/mL) was administered using a calf esophageal tube. Six days following the administration, calves in group C were inoculated with *M. haemolytica* (10 mL per calf, ~3 × 10^9^ CFU/mL) by trans-trachea injection using a catheter to induce BRD according to the protocol described in our publication [76]. In the subsequent days, the calves were monitored for signs of BRD, such as elevated body temperature, eye and nasal discharges, ear droop or head tilting, cough, and changes in breathing, eating, and ambulation. Eight days after trans-tracheal inoculation of *M. haemolytica,* calves in groups B and C were subcutaneously injected with a single dose of danofloxacin (ADVOCIN™, *danofloxacin mesylate*, Pfizer Animal Health; 8 mg/kg body weight) in the neck. Fecal samples were collected to a 50 mL screw-cap tube directly from the rectum from all study calves on days 2, 6, 14, 18, 19, 22, and 28; four times before the treatment (i.e., danofloxacin injection) and three times after the treatment. The calves were euthanized with a penetrating captive bolt gun at the end of the study, as per AVMA Guidelines on Euthanasia [77].

### 4.2. DNA Extraction, Library Preparation, and Sequencing

**16S rRNA**: To determine the effects of danofloxacin on gut microbiota, 16S rRNA analysis was conducted. DNA extractions were performed following ZymoBIOMICS™ protocol from 210 fecal samples (10 calves per group and seven sampling time points). The V4-V5 hypervariable regions of the bacterial 16S rRNA gene were amplified using a universal 16S forward primer (515F: GTGYCAGCMGCCGCGGTAA) and a reverse primer (926R: CCGYCAATTYMTTTRAGTTT). Briefly, the fecal samples were thawed at room temperature for approximately 30 min. From each sample, 200 mg of feces was transferred to a 2 mL ZR BashingBead™ lysis tube and mixed with 250 µL deionized sterile water, 750 µL lysis solution, and 50 µL proteinase K. The samples were processed by a bead beater for 10 min followed by incubation for at least 30 min in a water bath at 55 °C. Then, the lysis tubes were centrifuged in a microcentrifuge at 10,000 × *g* for 3 min. The supernatant was harvested to columns and then washed with DNA Wash Buffer 1 and 2. The final product was eluted with DNase/RNase free water, and the concentration of eluted DNA was measured first by NanoDrop 3300 Fluorospectrophotometer (Nanodrop technologies, USA) and confirmed by Qubit fluorometer (Invitrogen). Following normalization of all the DNA extracts, they were transferred to 96-wells plates, and submitted for sequencing to the DNA Facility of Iowa State University. The Earth Microbiome Project protocol was followed for sequencing on the Illumina MiSeq platform (2 × 250 paired-ends) in a single flow cell lane. For control, two community standards were used.

**Shotgun sequencing:** To assess the effects of danofloxacin on gut microbial resistome, shotgun and ProxiMeta Hi-C metagenomics were performed. For shotgun library preparation, four samples (pooled two pre-treatment samples and pooled two post-treatment samples from calves in Group B) were selected based on the results of 16S analysis. Samples collected right before danofloxacin injection (i.e., Day 18) and four days later (i.e., Day 22) were used for this purpose. Like in the case of 16S rRNA gene, whole-genome DNA was extracted from the fecal samples according to ZymoBIOMICS™ instructions. The whole-genome extracts were submitted to the DNA Facility of Iowa State University, where a single flow cell lane Illumina HiSeq platform (2 × 150 bp) was used for sequencing.

**Metagenomic Hi-C ProxiMeta DNA extraction and library preparation:** The same samples used for the whole-genome shotgun DNA extraction were used for the Metagenomic ProxiMeta sequencing. The Hi-C library was created using a Phase Genomics (Seattle, WA, USA) ProxiMeta Hi-C Microbiome Kit, which is a commercially available version of the Hi-C protocol. Briefly, 100 mg of the fecal sample was washed with TBS, and then genetic material (both chromosomal and non-chromosomal) were crosslinked in vivo while the bacterial cells were still intact using a formaldehyde solution, simultaneously digested using restriction enzymes Sau3AI and MlucI. The genetic materials were proximity ligated with biotinylated nucleotides to create chimeric molecules composed of fragments from different regions of genomes that were physically proximal in vivo according to the manufacturer’s instructions for the kit. The chance of inter-cellular interactions of genetic materials was negligible. Molecules were pulled down with streptavidin beads and processed into an Illumina-compatible sequencing library according to the protocol. Sequencing was performed on an Illumina HiSeq instrument (2 × 150 bp). The bioinformatic analyses were described in our recent publication [78].

**Quantitative real-time PCR** (**qPCR**): To assess the dynamics of antimicrobial resistance genes following danofloxacin injection in the fecal samples, we ran qPCR using primers previously designed by Looft et al. [79] presented in Table 5. The target resistance genes were selected based on the metagenomic Hi-C results; accordingly, primers were ordered from the ISU DNA facility for *tet*W, *tet*O, *tet*X, *erm*B, and *erm*F. The DNA extracts (described above) from fecal samples collected on Day 18 for the pre-treatment and Day 28 for the post-treatment, were pooled together for each group separately. PCR assays were run using the SsoAdvanced™ Universal SYBR^®^ Green Supermix (Bio-Rad, Hercules, CA, USA) and the CFX Maestro™ Real-Time PCR detection system (Bio-Rad). Dilutions of DNA template for both standards (16S and target genes) and all unknowns were run in triplicate with reaction volumes of 10 μL. Amplification of DNA occurred with 35 cycles of denaturation at 95 °C for 10 s and then annealing for each primer pair at 60 °C for 30 s. Both standard curves (16S and target genes) were experimentally validated to have high efficiency > 90% of amplification and precision R^2^ ~ 0.98 prior the analysis. Relative expression was normalized using 16S detection levels. The relative fold change of detection between the control and treatment groups (B and C) were calculated using the ISU Gallup Method Equation [80]. Statistical analysis was performed using Dunn pair-wise comparison test in Rstudio to determine significance changes in ARG levels between the pre-and post-treatment pooled samples. An adjusted *p*-value of < 0.05 was considered significant.

### 4.3. Bioinformatics and Data Analysis of 16S Data

Bioinformatic analyses of the 16S rRNA sequence data were performed using the QIIME 1.9 and QIIME 2 pipeline. Alpha and beta diversity metrics were computed to evaluate the effects of the antibiotic on the richness and diversity of gut microbiota. The richness was estimated by using observed OTUs, and the evenness using the Shannon index. A non-metric multidimensional scaling plot (distance = bray, k = 3) was developed using the Vegan R package to visualize the clustering difference between pre- and post-treatment microbial communities.

We used R statistical software for statistical analyses and plotting of graphs. The Kruskal–Wallis test was used to compare the relative abundances of bacterial taxa among the three calf groups and Dunn test for pair-wise comparisons. Similarly, pre- and post-treatment bacterial OTUs were compared by the Wilcoxon signed-rank test. The association of the relative abundance of genus *Campylobacter* with other genera was explored using the Spearman rank correlation test, and multiple testing correction with a false discovery rate was performed [5]. A *p*-value < 0.05 was considered significant for all analyses, and for the false discovery rate, *p* adjusted < 0.05 was used unless specified otherwise.

#### Prediction Models Using Random Forest Models

In total, we obtained 298 genera in our samples; 70 of them with very low relative abundance (zero-inflated) were excluded from further analyses. Among 228 genera, important genera that might have a significant association with pre- and post-treatment status and the compositions of *Campylobacter* were selected using the Boruta algorithm. For the building of random forest models, the data were randomly divided into a training sub-data (70%) and a test sub-data (30%). The models were first trained using the training sub-data, and then its fitness was evaluated using the test sub-data. The genera selected by the Boruta algorithm were fed to the random forest models to rank the genera based their predictive importance and presented in graphs. The Boruta algorithm and random forest models have been used in the microbiota studies to select and rank important variables from high-dimensional data [52,53,54]. In this study, we used these methods to identify bacterial taxa that are capable of classifying the microbial communities into pre- and post-treatment, as well as predicting the relative abundance of *Campylobacter.* For the latter case, further analyses were performed by subsetting data as pre-treatment, post-treatment, and control to check if the treatment might have affected the association patterns between genera.

## 5. Conclusions

In summary, our data indicate that subcutaneous administration of danofloxacin significantly altered the compositions and structures of the fecal microbiota in calves. However, not all bacterial taxa were affected alike by the treatment. Several of them showed a reduction in relative abundances, a few other taxa were enriched, and many of them remained the same. The relative abundance of the genus with public health importance, *Campylobacter*, abruptly reduced 24 h after the treatment but was resilient after three days, which may imply that this bacterium developed resistance to danofloxacin and, thus, it entails a risk to the public health. The novel ProxiMeta Hi-C method revealed that several types of ARGs that confer resistance to commonly used antibiotics in cattle were enriched by danofloxacin. Additionally, using this method, the presumptive reservoirs of resistance genes have been predicted, and this will help to make further investigations to acquire a deeper understanding of the way resistance genes spread within microbial communities in general and the way they transmit to bacterial species that have public health significance in particular.

## Figures and Tables

**Figure 1 microorganisms-09-02023-f001:**
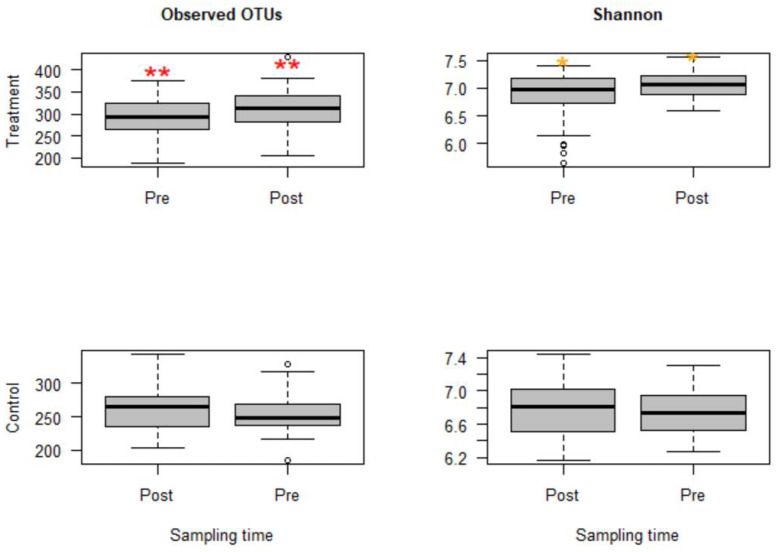
Boxplots for alpha diversity metrics comparing pre- and post-treatment samples in the treatment group (**top**) and control group (**bottom**). The treatment group received a single subcutaneous injection of danofloxacin. Asterisks show significant differences (* *p* < 0.05, ** *p* < 0.01) between pre- and post-treatment microbial communities for the respective group. Note: the control group was divided into pre- and post-treatment to compare with the treatment group in a temporal manner; otherwise, the calves in the control group did not receive any antibiotic. The observed OTUs and Shannon index do not have units, but they measure microbial richness and evenness based on OTU counts and abundance.

**Figure 2 microorganisms-09-02023-f002:**
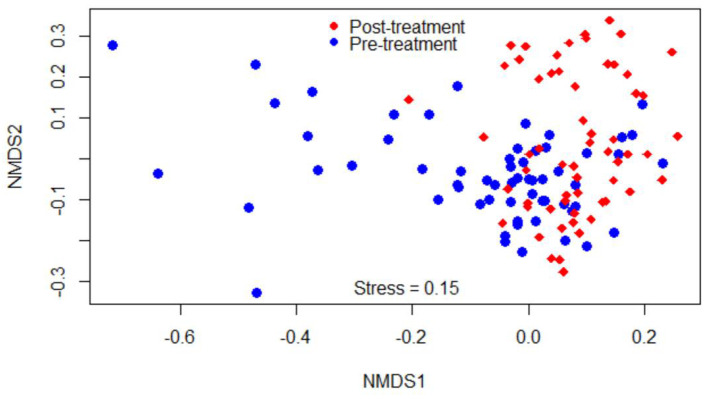
Non-metric multidimensional scaling (NMDS) of the Bray-Curtis dissimilarity index showing clustering difference between pre- and post-treatment microbial communities, blue and red dots, respectively.

**Figure 3 microorganisms-09-02023-f003:**
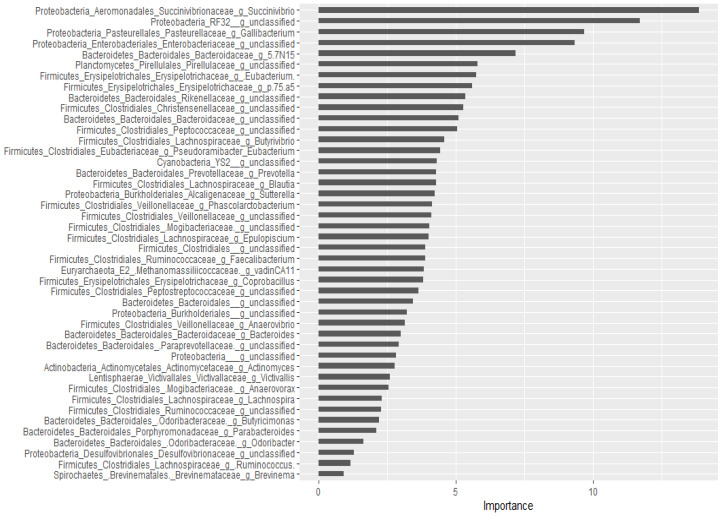
Bacterial genera that classify pre- and post-treatment communities based on the Boruta important variable selection algorithm and ranking by the random forest model. The length of the horizontal bars corresponds to its predictive importance.

**Figure 4 microorganisms-09-02023-f004:**
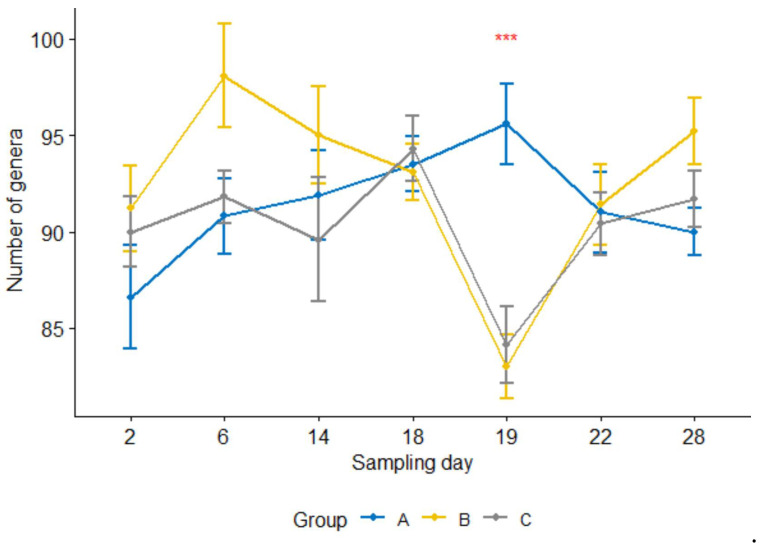
Number of genera per sample for three calf groups by sampling time point. Genus count at sampling Day 19 were significantly lower in both treatment groups B and C than in the control group A. *C. jejuni* cocktail, *M. haemolytica*, and danofloxacin were administered on Days 4, 10 and 18, respectively. *** *p* = 0.001.

**Figure 5 microorganisms-09-02023-f005:**
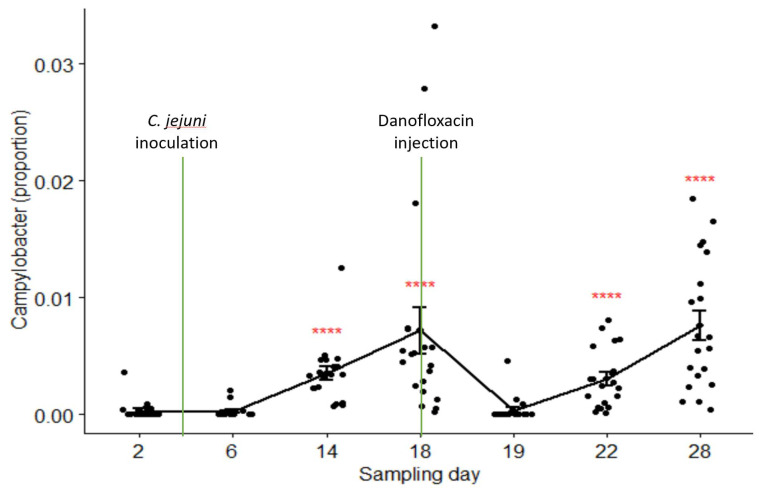
Comparisons of the relative abundance of *Campylobacter* in samples collected from the calves in the combined treatment groups (B and C) by sampling days. Danofloxacin was injected on Day 18 right after fecal sample collection. **** relative abundance of *Campylobacter* on Day 19 was significantly different from those on the sampling Day 14, Day 18, Day 22, and Day 28. The abundance rates were significantly different from each other on sampling Day 22 and Day 28 (*p* = 0.000).

**Figure 6 microorganisms-09-02023-f006:**
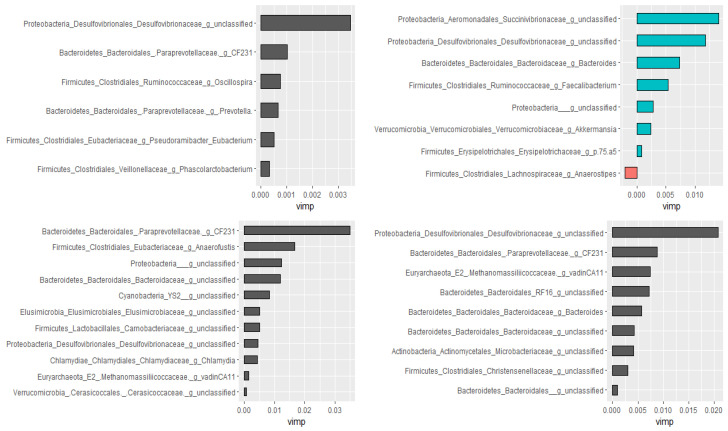
Genera associated with the relative abundance of *Campylobacter* selected by the Boruta algorithm and ranked by the random forest models. Control (**top left**), pre-treatment (**top right**), post-treatment (**bottom left**), and all data (**bottom right**). The length of the bar is proportional to the predictive importance of the genera; vimp—variable importance.

**Table 1 microorganisms-09-02023-t001:** Comparisons of the relative abundance of bacterial genera between fecal samples collected before and after danofloxacin injection in the treatment group by the Wilcoxon signed-rank test. Only significantly altered genera were shown in the table.

Genus	Relative Abundance (%)		*p*	Genus	Relative Abundance (%)		*p*
	Pre	Post			Pre	Post	
Succinivibrio	0.38	0.05	0.000	Erysipelotrichales_RFN20	0.04	0.03	0.000
Streptophyta_5-7N15	4.12	8.31	0.000	Desulfovibrio	0.51	0.27	0.000
Enterobacteriaceae_unclassified	0.08	0.01	0.000	Elusimicrobiaceae_unclassified	0.12	0.04	0.000
Peptococcaceae_unclassified	0.13	0.25	0.000	Bacteroidaceae_unclassified	1.53	2.54	0.000
Alphaproteobacteria_unclassified	0.38	0.14	0.000	Synergistes	0.03	0.01	0.000
Eubacterium	0.13	0.03	0.000	Methanosphaera	0.35	0.22	0.001
Coprobacillus	0.01	0.00	0.000	Sutterella	0.61	0.24	0.001
Erysipelotrichales_p-75-a5	0.10	0.00	0.000	Pseudoramibacter	0.05	0.02	0.001
Prevotella	2.29	0.93	0.000	Rikenellaceae_unclassified	5.57	3.44	0.003
Blautia	0.27	0.05	0.000	Epulopiscium	0.04	0.08	0.004
[Paraprevotellaceae]_unclassified	0.04	0.00	0.000	Treponema	1.98	1.45	0.005
Faecalibacterium	0.15	0.00	0.000	Clostridiales_unclassified	9.72	8.65	0.005
Phascolarctobacterium	0.51	0.36	0.000	[Barnesiellaceae]_unclassified	0.19	0.32	0.008
Anaerovibrio	0.10	0.03	0.000	Bifidobacterium	0.37	0.12	0.013
Mogibacteriaceae_unclassified	0.69	0.50	0.000	Desulfovibrionaceae_unclassified	0.20	0.28	0.014
Butyrivibrio	0.09	0.16	0.000	Akkermansia	2.94	4.72	0.018
Cyanobacteria_unclassified	1.16	0.75	0.000	Elusimicrobium	0.22	0.29	0.022
Veillonellaceae_unclassified	0.18	0.13	0.000	Peptostreptococcaceae_unclassified	1.61	2.33	0.023
Ruminococcaceae_unclassified	27.17	31.87	0.000	Bacteroides	1.83	0.23	0.042
Pirellulaceae_unclassified	0.05	0.14	0.000	Ruminobacter	0.01	0.02	0.046

**Table 2 microorganisms-09-02023-t002:** Summary of antimicrobial resistance genes in pooled fecal samples before and after danofloxacin injection in calves identified by Metagenomic Hi-C.

Antibiotic Class	Resistance Gene	Number of Hits		Number of Hosts	
		Pre-trt ^a^ (165 *, 38 **)	Post-trt ^b^ (200, 64)	Pre-trt (165, 38)	Post-trt (200, 64)
Aminoglycoside	*aac*6	0	3	0	3
	*aph*2	61	58	14	16
	*aph*3	51	55	11	13
	*ant*6	75	84	22	29
	*ant*9	39	189	12	23
	*sat*	51	56	11	12
Beta-lactams	*aci*	1	4	1	2
	*cf*X	5	8	5	3
Macrolides	*erm*B	8	0	3	0
	*erm*F	0	1	0	1
	*erm*G	24	5	3	3
	*erm*Q	1	0	1	0
	*mef*E	3	1	2	1
Phenicol	*cf*R	15	24	3	14
Tetracyclines	*tet*32	20	16	9	7
	*tet*40	112	1484	44	91
	*tet*44	15	6	10	5
	*tet*A	8	4	5	3
	*tet*B	1	0	1	0
	*tet*L	0	69	0	5
	*tet*O	78	75	27	21
	*tet*Q	25	483	10	19
	*tet*W	178	2836	79	105
	*tet*X	0	1	0	1

^a^ Pre-treatment samples, pooled individual samples collected right before danofloxacin injection (Day 18); ^b^ Post-treatment samples, pooled individual samples collected four days after danofloxacin injection (Day 22). * Total number of clusters in the pooled samples, and ** number of clusters with ≥ 80% complete genomes.

**Table 3 microorganisms-09-02023-t003:** Summary of phyla associated with antimicrobial resistance genes detected in pooled fecal samples before and after danofloxacin administration.

Phylum	Pre-Treatment	Post-Treatment
*Actinobacteria*	ermB, ermG, tetW	ant9, tet40, tetQ, tetW
*Bacteroidetes*	tetQ	aph2, aph3, ant6, cfX, ermF, ermG, mefE, cfR, tet40, tetQ, tetW
*Euryarchaeota*	ant9, tetA, tetO	No ARGs
*Firmicutes*	aph2, aph3, ant6, ant9, sat, cfX, ermB, cfR, tet32, tet40, tet44, tetA, tetB, tetO, tetQ, tetW	aac6, aph2, aph3, ant6, ant9, sat, ermG, cfR, tet32, tet40, tetA, tetL, tetO, tetQ, tetW
*Proteobacteria*	aph2, aph3, ant6, ant9, sat, tet32, tet40, tetO, tetW	aac6, aph2, aph3, ant6, ant9, cfR, tet40, tetL, tetQ, tetW
*Spirochaetes*	NA	ant6, sat, tet40, tet44, tetW
*Tenericutes*	tet40	No ARGs
*Verrucomicrobia*	tet40	ant6, sat, tet40, tetW

**Table 4 microorganisms-09-02023-t004:** Changes in the quantity of selected antimicrobial resistance genes in pooled fecal samples following danofloxacin treatment measured by qPCR.

	Change	Control	Group B	Group C
*tet*W	Mean (log2 *)	−0.21	0.58	0.34
	SD	0.017	0.065	0.015
	*p* value **	NA	0.034 ^a^	0.272
*tet*O	Mean	−0.01	−0.15	−0.32
	SD	0.024	0.088	0.173
	*p* value	NA	0.272	0.0338 ^c^
*tet*X	Mean	−0.31	−0.92	0.02
	SD	0.081	0.058	0.039
	*p* value	NA	0.267	0.264
*erm*B	Mean	0.52	−0.76	−0.15
	SD	0.081	0.064	0.174
	*p* value	NA	0.022 ^b^	0.359
*erm*F	Mean	−0.68	−0.44	0.19
	SD	0.054	0.095	0.173
	*p* value	NA	0.359	0.022 ^d^

* The values obtained using Gallop method analysis was log2 transformed to compare the fold changes in gene abundance between calf groups. ** An adjusted *p*-value obtained from pairwise comparisons using the Dunn test. ^a^
*tet*W increased and ^b^
*erm*B decreased significantly in the group B post-treatment pooled sample compared to the control group post-treatment sample. ^c^
*tet*O decreased and ^d^
*erm*F increased significantly in group C compared to the control group.

**Table 5 microorganisms-09-02023-t005:** Primers used in quantitative real-time PCR tests to assess the dynamics of selected antimicrobial resistance genes in pooled fecal samples from calves.

ARGs	Forward Primer	Reverse Primer	Reference
*erm*B	TGAAAGCCATGCGTCTGACA	CCCTAGTGTTCGGTGAATATCCA	[79]
*erm*F	TTTCAAAGTGGTGTCAAATATTCCTT	GGACAATGGAACCTCCCAGAA	
*tet*O	ATGTGGATACTACAACGCATGAGATT	TGCCTCCACATGATATTTTTCCT	
*tet*W	TCCTTCCAGTGGCACAGATGT	GCCCCATCTAAAACAGCCAAA	
*tet*X	AAATTTGTTACCGACACGGAAGTT	CATAGCTGAAAAAATCCAGGACAGTT	

## Data Availability

Data are available upon request.

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
