# Peer review of "Danofloxacin Treatment Alters the Diversity and Resistome Profile of Gut Microbiota in Calves"

_microorganisms, 2021, doi:10.3390/microorganisms9102023_

Round 1

Reviewer 1 Report

Dear Authors,
I believe that the study you have conducted is very interesting and above all, it provides a large amount of data in the context of the topic addressed.
Precisely for this reason, I believe that by better organizing the paper in its structure, scholars interested in reading and understanding it would be easier.
Therefore, in order not to be offensive, by strongly supporting all the work that Veterinarians carry out inside and outside the academic world, I report my suggestions:

Line 33: Please remove "bovine calves", as it does not seem necessary to specify further what animal species you are dealing with.

Lines 103-110: I suggest removing all of this period which is strictly related to materials and methods.

Line 281: the subsection heading should be 2.3. Please change.

Line 320: the subsection number should be 2.4. Please change.

Line 336 below (Discussion): to make the discussion more structured and easier to follow, I recommend dividing it into subsections consistently with the results.

Author Response

Thank you for reviewing our manuscript and providing excellent suggestions.

“The authors could improve the impact of the manuscript by enriching their discussion on antimicrobial resistance (AMR) and its importance to public health from a One Health perspective. Moreover, AMR is serious endemic public health problem, much quieter than the COVID-19 pandemic, but as crucial in the context of the United Nations 2030 Agenda specifically for achieving Sustainable Development Goal 3 (SDG3-Good Health).”

We really appreciate your thoughtful suggestion to improve our paper and strongly agree that it could be enriched by discussing our findings in the context of the “One Health” perspective. The following improvements have been made based on your kind suggestion:

Introduction: lines 45-48: “This situation demands a clear understanding of variables involving in the emergence and maintenance of antimicrobial-resistant bacterial strains and genes on the livestock farms and a “One Health” approach to curbing their spread to environments and humans.”

Discussion: lines 357-361: “The emergence of resistant strains and genes in livestock is worrisome since they shed them with feces and contaminate the farm environment, which may eventually reach the human food chain or drinking water. This necessitates capturing “One health” paradigm to study AMR and minimizing its economic and public health impacts [7, 8].”

Reviewer 2 Report

Well-written and interesting manuscript on a hot topic.

The authors could improve the impact of the manuscript by enriching their discussion on antimicrobial resistance (AMR) and its importance to public health from a One Health perspective. Moreover, AMR is serious endemic public health problem, much quieter than the COVID-19 pandemia, but as crucial in the context of the United Nations 2030 Agenda specifically for achieving Sustainable Development Goal 3 (SDG3-Good Health).

Author Response

We are impressed with your thoughtful comments and suggestions on our manuscript. Thank you for your time and inputs. Below, we have kindly addressed your comments and questions.

1. Authors study the effect of subcutaneous injection, however, different routes of injections may vary the effect on the gut microbiome. What is the author’s opinion about this? Please discuss.

We appreciate your suggestions and question. It is a legit question. The outcomes of parenteral and oral administration of antibiotics, in general, may follow different patterns in affecting gut bacteria. In this paper, we addressed the other rout such as intramuscular (lines 89-92 and 383-387). The following statements comparing oral and IM administration is added in the discussion

Lines 385-388. “The same study revealed that both orally and parenterally administered enrofloxacin induced development of AMR in selected bacterial species at different levels in pigs [42]; however, the difference in the impacts of fluoroquinolone drugs administered via oral and parenteral routes on gut microbiota needs to be investigated in calves.”

2. Authors studied the gut microbial composition using fecal samples. Did the authors concern for anaerobic microbial communities also? If not authors should mention it clearly. Please discuss.

Thank you for this excellent question and suggestion. In this study, we assessed the microbial composition based in 16S rRNA gene, which is a well-established metagenomic approach. It is a gene-based method that does not require the viability of the organisms. Thus, we do not have much concern about anaerobic microbial communities. However, we agree with the reviewer that this method, like other microbiology methods, have limitations. Nevertheless, given that two thirds of the gut/fecal microorganisms are not culturable, 16S approach is one of the best metagenomic methods that we have to study fecal microbial compositions.

3. What are the units for Y-axis? Why there is a difference in the Y-axis units in OUT’s and Shannon? Please clarify and all the details should be mentioned in the legend.

The alpha diversity metrics measure different aspects of microbial community. The observed OTUs measures the richness (i.e., number of species) of the community, while the Shannon index represents microbial evenness. Both metrics do not have a unit, but their computation using different mathematical equations resulted in different figures. Thank you for your suggestions, and accordingly we added the following statement in the legend.

Lines 165-167: The observed OTUs and Shannon index do not have units, but they measure microbial richness and evenness based on OTU counts and abundance.

4. Authors, should collect and isolate some bacterial species on different media to compare the microbial communities before and after antibiotic injection, in addition to meta-genomic study alone.

We appreciate your comments. Most of gut bacterial species are difficult to grow on artificial media; thus, studying their phenotypic resistant status is hardly possible. However, as the reviewer suggested representative bacterial species can be cultured and ARG dynamics can be evaluated. In this light, another manuscript under preparation from the same study presents the change in antimicrobial resistant profile of Campylobacter species following the treatment and it will be submitted to this journal in a few weeks.

5. Authors should also do some antimicrobial resistance assay to confirm the resistome profile.

Thank you for this suggestion. We strongly believe that phenotypic AMR experiments are confirmatory for molecular based studies. A separate paper that contains results based on antimicrobial assays is under preparation.

Reviewer 3 Report

  1. Authors study the effect of subcutaneous injection, however, different routes of injections may vary the effect on the gut microbiome. What is the author’s opinion about this? Please discuss.
  2. Authors studied the gut microbial composition using fecal samples. Did the authors concern for anaerobic microbial communities also? If not authors should mention it clearly. Please discuss.
  3. What are the units for Y-axis? Why there is a difference in the Y-axis units in OUT’s and Shannon? Please clarify and all the details should be mentioned in the legend.
  4. Authors, should collect and isolate some bacterial species on different media to compare the microbial communities before and after antibiotic injection, in addition to meta-genomic study alone.
  5. Authors should also do some antimicrobial resistance assay to confirm the resistome profile.

Author Response

(The authors gave the same response as above.)

Round 2

Reviewer 3 Report

The authors replied to the reviewer's concerns.